# Quick Back-Translation for Unsupervised Machine Translation

**Benjamin Brimacombe**
Columbia University
bb2808@columbia.edu

**Jiawei Zhou**
Harvard University
jzhou02@g.harvard.edu

## Abstract

The field of unsupervised machine translation has seen significant advancement from the marriage of the Transformer and the back-translation algorithm. The Transformer is a powerful generative model, and back-translation leverages Transformer's high-quality translations for iterative self-improvement. However, the Transformer is encumbered by the run-time of autoregressive inference during back-translation, and back-translation is limited by a lack of synthetic data efficiency. We propose a two-for-one improvement to Transformer back-translation: Quick Back-Translation (QBT). QBT re-purposes the encoder as a generative model, and uses encoder-generated sequences to train the decoder in conjunction with the original autoregressive back-translation step, improving data throughput and utilization. Experiments on various WMT benchmarks demonstrate that a relatively small number of refining steps of QBT improve current unsupervised machine translation models, and that QBT dramatically outperforms standard back-translation only method in terms of training efficiency for comparable translation qualities.

## 1 Introduction

The capabilities of neural machine translation (MT) models have seen considerable advancement in recent years (Vaswani et al., 2017; Hassan et al., 2018; Nguyen et al., 2021). However, paired translation data for supervision is often difficult to procure without specialized expertise, especially for low-resource languages. Proposed originally in the statistical machine translation regime (Ravi and Knight, 2011; Klementiev et al., 2012), unsupervised machine translation (UMT) aligns languages without parallel data by leveraging vocabulary-level statistics or abstract language representations. Modern UMT approaches combine self-supervised learning and statistical methods (Sennrich et al., 2016a; He et al., 2016; Lample et al.,

2018a; Artetxe et al., 2018), and more recently pair generative language modeling with iterative back-translation (Conneau and Lample, 2019; Song et al., 2019; Nguyen et al., 2021). Iterative back-translation (BT) creates synthetic translation pairs from monolingual corpora, and uses them to train neural MT models with a reconstruction loss (Sennrich et al., 2016a; Conneau and Lample, 2019). Models trained with iterative back-translation hold the state-of-the-art (SOTA) performance on a variety of UMT scenarios (Conneau and Lample, 2019; Song et al., 2019; Roziere et al., 2020; Nguyen et al., 2021). Recent work suggests that increased data diversity during back-translation improves translation performance, and further increasing diversity remains an open challenge (Wang et al., 2019; Nguyen et al., 2021).

Back-translation involves massive amounts of sequence generation, which usually follow an autoregressive (AR) inference procedure. In AR inference, tokens are generated one step at a time, which can not be parallelized (Vaswani et al., 2017; Kasai et al., 2020). Non-autoregressive (NAR) Transformer models have been proposed as a method to improve inference speed for translation models, in which sequence transduction is done in parallel (Gu et al., 2018; Wei et al., 2019; Zhou and Keung, 2020).

In this paper, we address both autoregressive inference and data diversification limitations with a novel, framework for UMT, called Quick Back-Translation (QBT). With QBT, the Transformer encoder is trained as a bidirectional generative translation model. In what we call the Encoder Back-Translation (EBT) step, the encoder back-translates its own synthetic outputs, training as an independent UMT model. The encoder-synthesized translations are then leveraged as synthetic signal for the decoder, providing increased data diversity in what we call an Encoder Back-Translated Distillation (EBTD) step. The Transformer architecture is kept

intact under Quick Back-Translation, allowing the new optimization procedures to supplement or enhance the original back-translation objective. The QBT methods add more diverse training signals for UMT by dual usage of the encoder both as a module inside the Transformer and as a standalone NAR translation model, which offer a significant speedup on top of standard BT. We conduct experiments on multiple translation benchmarks of different resources, and QBT is able to achieve UMT qualities on par with previous SOTA but with high efficiency in run-time, especially for long sequences.

## 2 Background

**Unsupervised Machine Translation** For many language pairs there do not exist sufficient examples to train a supervised translation model. It has been proposed to leverage monolingual data only to learn a map between languages, called Unsupervised Machine Translation (UMT). Originally proposed as a probabilistic decipherment problem (Ravi and Knight, 2011; Klementiev et al., 2012), modern UMT now leverages representation learning (Lample et al., 2018b) and pre-trained generative models (Artetxe et al., 2018; Conneau and Lample, 2019; Nguyen et al., 2021). Generative-model-based UMT techniques rely on language model pre-training and self-supervised learning (Radford et al., 2018). Common pre-training objectives include Masked Language Modeling (MLM), Causal Language Modeling (CLM), Denoising Autoencoding (DAE), and the Masked Sequence to Sequence (MASS) objective (Vaswani et al., 2017; Devlin et al., 2019; Conneau and Lample, 2019; Song et al., 2019), which are used for initializing UMT models.

**Back Translation** Back-Translation (BT) refers to a model or set of models that recursively train using reverse synthetic translations. Originally proposed to supplement parallel data during supervised MT training (Sennrich et al., 2016a), BT has been extended to multi-agent scenarios (He et al., 2016), and fully unsupervised translation in the statistical machine translation regime (Artetxe et al., 2018) and recently with language modeling (Lample et al., 2018a; Conneau and Lample, 2019; Song et al., 2019). Most recently, the Cross-model Back-translated Distillation (CBD) method (Nguyen et al., 2021) used back-translation between models pre-trained with independent back-

translation, all in the fully unsupervised scenario. They find that more synthetic data diversity improves even high quality UMT models.

**Non-autoregressive Generation** The Transformer uses an autoregressive (AR) decoder, in which each token output is conditioned on previously seen tokens. Inference consists of multiple generation steps increasing with the sequence length. BT with Transformer relies on this inefficient step, and hence is often conducted on short sequences, using large amounts of compute (Edunov et al., 2018; Song et al., 2019; Nguyen et al., 2021). Non-autoregressive (NAR) models (Gu et al., 2018; Wei et al., 2019) have been proposed as a solution to improve inference speed for MT models. NAR MT conducts full sequence transduction in parallel. However, the parallel decoding requires an assumption of conditional independence between sequence tokens, which is believed to hinder model performance due to the underlying multi-modal distribution of natural language (Gu et al., 2018; Guo et al., 2020; Song et al., 2021). A number of NAR techniques leverage knowledge distillation with an AR teacher to improve training signal and mitigate the independent factorization problem. Common NAR Transformer models usually modify the decoder computation by removing the AR self-attention mask and feeding sequential inputs all at once (Gu et al., 2018; Zhou and Keung, 2020), whereas in our approach we directly re-purpose the Transformer encoder for the NAR generation in the context of BT to improve UMT learning efficiency.

We propose a counter-intuitive flip of the distillation from strong to weak models, and use a weak, bidirectional encoder to improve a strong, AR decoder. Furthermore, our weak NAR model is hiding in plain sight in the Transformer architecture. Our architecture pushes the synthetic data diversity findings from Nguyen et al. (2021) to an extreme in which the distillation model is a highly limited NAR architecture. We find that this is sufficient to improve upon the BT-only algorithm, improving training speed and translation performance in a variety of experimental scenarios.

**Large Language Models for UMT** The advent of large language models (LLMs) has greatly improved many tasks with zero-shot and few-shot in-context learning. However, UMT is still relevant, especially when the resource is limited in terms of computation and language data. It is ar-

guable whether some LLMs can perform MT in an unsupervised manner, as the massive scale of web data might already contain paired language corpora (Magar and Schwartz, 2022). Nevertheless, in-context back-translation with pseudo parallel examples has been shown to outperform other prompting methods for language translation (Zhang et al., 2023). In addition, back-translation and UMT techniques remain prominent for the leading large models such as mBART (Liu et al., 2020), and similar full encoder-decoder architectures are still employed such as mT5 (Xue et al., 2021) and FSMT (Ng et al., 2019). Our work aims to improve the efficiency of BT algorithms with encoder-decoder Transformers, orthogonal to the development of versatile LLMs.

## 3 Preliminaries

Let $\mathcal{D}_s$ and $\mathcal{D}_t$ be the monolingual corpus of sentences in a source domain $s$ and a target domain $t$, respectively, and they are not aligned. Given a source sentence $x = (x_1, x_2, \ldots, x_m) \sim \mathcal{D}_s$ with length $m$, UMT aims to learn a conditional generation model $p_\theta(y|x)$ with parameters $\theta$ so that $y = (y_1, y_2, \ldots, y_n) \sim \mathcal{D}_t$ with length $n$ is closest to the true translation for $x$ after decoding.

**Transformer Model** The Transformer (Vaswani et al., 2017) is a sequence-to-sequence model with an encoder-decoder architecture built with attention mechanisms. Given the source sentence $x$, it models the conditional target distribution via

$$h_e = \text{Enc}(\text{Emb}(x; W_e); \theta_e)$$
$$h_{d,t} = \text{Dec}(\text{Emb}(y_{<t}; W_d), h_e; \theta_d) \quad (1)$$
$$p_\theta(y|x) \propto \text{Softmax}(\text{Linear}(h_d; W_{do}))$$

where $h_e$ and $h_d$ are hidden states out of encoder and decoder computations, $t$ denotes generation steps, $W_e$ and $W_d$ are embedding matrices for encoder and decoder inputs, $W_{do}$ is the weight matrix for decoder output[1] to map $h_d$ to the vocabulary size, and $\theta_e$ and $\theta_d$ are parameters for the encoder and decoder layers, respectively. Each layer of the encoder and decoder is composed of self-attention modules followed by feedforward networks,[2] and the decoder layer also consists of additional cross-attention modules attending to encoder outputs $h_e$.

---

[1] We ignore the bias terms in linear layers hereafter.
[2] There are also positional embeddings, layernorm and residual connections which we skip discussion for simplicity.

There is a major difference between the self-attention computation of the encoder and decoder in sequence generation. The encoder self-attention allows each token representation to attend to all the other tokens before and after it, thus is bidirectional. For the decoder, as the future tokens are unknown before being generated sequentially, each token representation at $t$ can only attend to the previous tokens $y_{<t}$, which is autoregressive. During conditional generation such as MT, the encoder only needs to be run once with full parallelization on $x$, and the decoder needs to be run $n$ steps for every $t$, making it a bottleneck for generation speed.

**Back-Translation** By translating in the backward direction from target to source, BT (Sennrich et al., 2016a) provides synthetic parallel corpus for data augmentation used by many UMT approaches. In particular, BT generates $x \sim \mathcal{D}_s$ from a reverse model $p_\phi(x|y)$ based on a target sample $y \sim \mathcal{D}_t$, and then uses $(x, y)$ as pseudo pairs for supervision for learning the source to target model $p_\theta(y|x)$. For UMT, BT is mostly used in an iterative fashion (Hoang et al., 2018), switching between the source language $s$ and the target language $t$ to jointly improve the language alignments and translation qualities (Lample et al., 2018a). In our setup we also tie both translation directions with the same model, essentially making $\phi = \theta$.

## 4 Quick Back-Translation

Iterative BT with Transformer relies on sequential generation of translations from the decoder, which may not be parallelizable due to its autoregressive design. We propose Quick Back-Translation (QBT), a non-autoregressive (NAR) generation approach to provide increased synthetic data and reduced inference time during BT. In contrast to common NAR Transformer designs, we directly repurpose the Transformer encoder as an NAR generation model without any modifications. In particular, the back-translation from target $y$ to source $x$ is obtained via

$$h_e = \text{Enc}(\text{Emb}(y; W_e); \theta_e)$$
$$p_\theta(x|y) \propto \text{Softmax}(\text{Linear}(h_e; W_{eo})) \quad (2)$$

and the argmax of the probability distribution is taken for each position independently in the sequence. Here the only new parameter is the output transformation matrix $W_{eo}$, which we tie with the encoder input embedding $W_{eo} = W_e$.

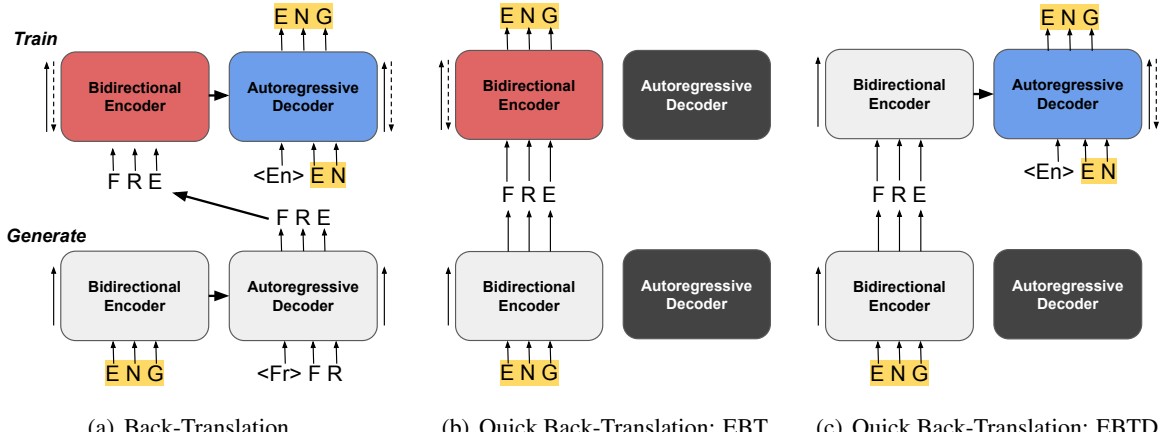

(a) Back-Translation     (b) Quick Back-Translation: EBT     (c) Quick Back-Translation: EBTD

Figure 1: (a) Back-Translation (b) Encoder Back-Translation (EBT) (c) Encoder Back-Translated Distillation (EBTD). BT, EBT and EBTD compose the Quick Back-Translation (QBT) procedure. The bottom row shows the generation step for back-translation, and the top row shows the training step with pseudo pairs. Light grey boxes indicate a forward step with no gradient updates. Dark grey boxes indicate a module that is not in use. An upward arrow next to the box indicate a forward run of the module, and a downward arrow indicate gradient and parameter updates.

This essentially shortcuts the computation from input to output by ignoring the decoder. It makes the translation NAR using only the bidirectional encoder whose computation is fully parallelizable with linear time complexity, with constant generation steps. Note however that output length is now confined to equal input length. An illustration of QBT in comparison with BT is presented in Figure 1. As an additional benefit to the fast run-time of encoder synthesis, QBT provides increased synthetic data diversity to the BT procedure, improving machine translation performance. In the following we describe different components of applying QBT for improving UMT models.

### 4.1 Encoder Back-Translation

Following generative language modeling or Transformer back-translation, the encoder is not a priori good at translating on its own. We propose the Encoder Back-Translation (EBT) phase to iteratively improve the encoder generation ability. As shown in Figure 1 (b), starting from samples $y \sim \mathcal{D}_t$, we first use encoder to generate $x \sim \mathcal{D}_s$, and then use the pairs $(x, y)$ to train the encoder, updating parameters $\{\theta_e, W_e\}$, with cross-entropy loss for the translation direction $x \rightarrow y$. The same is done for the reverse translation direction as well using the same encoder, and we alternative between these two directions similarly to Lample et al. (2018a). In this way we make the encoder a stripped down version of NAR encoder-

decoder models (Gu et al., 2018; Gu and Kong, 2021) trained for UMT. More details are provided in Algorithm 2 (see Appendix A).

### 4.2 Encoder Back-Translated Distillation

NAR generation still suffers from degeneration issues due to simplification of the sequential dependency structures (Gu et al., 2018; Ran et al., 2021; Guo et al., 2020). QBT focuses on accelerating the BT sampling process and increasing synthetic data diversity during UMT training. AR decoder based BT may be used in-conjunction with QBT to further improve translation quality. We propose Encoder Back-Translated Distillation (EBTD) to use encoder generated sequences as synthetic supervised data to train the decoder for generation. As illustrated in Figure 1 (c), we obtain paired sequences $(x, y)$ with fast NAR encoder by translating to $x$ from $y$, and then reversely feed $x$ to the encoder-decoder Transformer model to use $y$ as the target to update the decoder parameters $\{\theta_d, W_d\}$.[3] During the EBTD phase, the encoder parameters are frozen to preserve encoder translation quality. The EBTD algorithm details are outlined in Algorithm 3 (see Appendix A).

### 4.3 Initialization and Training Strategy

In practice, the QBT framework leverages BT, EBT and EBTD in synchrony to compose training. We

---

[3]We assume tied decoder output and input embeddings in our setup throughout the paper, i.e. $W_{do} = W_d$.

consider two configurations below, with different model initialization and training procedures applying QBT for different purposes.

**QBT-Synced**  In this scenario QBT is used to quickly boost translation performances of UMT models that have fully converged under standard BT training. The Transformer is initialized from a trained MT model. Then BT, EBT, and EBTD are run iteratively on the batch level, with EBT better aligning the Transformer encoder with translation objectives, EBTD adding diverse training signals from encoder generations for the Transformer decoder, and both leveraging the constant NAR generation time complexity. The algorithm description for QBT-Synced is shown in Algorithm 1.

**QBT-Staged**  Here we use QBT to supplement BT in training a UMT model from scratch. We initialize the parameters from a general pre-trained model that does not have any MT capability, and then run the following four training stages for UMT: training encoder with randomly paired source and target sentences, EBT training for the Transformer encoder, EBTD training for the Transformer decoder, and finally BT training for the full model parameters. In particular, the first stage is to prepare the encoder with basic language translation abilities for its direct QBT generation in later stages. We randomly sample $x \sim \mathcal{D}_s$ and $y \sim \mathcal{D}_t$ from two unrelated monolingual corpora to construct language pairs for warming up the encoder described in Equation (2).[4]  BT is still needed as encoder-generated translations are typically of lower quality than decoder-generated translations. The detailed algorithm for QBT-Staged is described in Algorithm 4 (see Appendix A due to space limit here).

## 5  Experimental Setup

**Data and Evaluation**  We evaluate our proposed methods on WMT News Crawl datasets.[5]  We prepare all of the monolingual data for English, French, German, and Romanian, up to year 2017. The datasets have sentence counts of 190M, 62M and 270M, 2.9M respectively. We encode data with Byte-Pair Encoding (BPE) (Sennrich et al., 2016b) with a dictionary of 60K sub-words provided by Conneau and Lample (2019). Each language pair: English and German, English and French, and En-

---

**Algorithm 1** Quick Back-Translation Synced (QBT-Synced): Given two monolingual corpora $\mathcal{D}_s$, $\mathcal{D}_t$, and an encoder-decoder Transformer model $\theta = \{\theta_e, W_e, \theta_d, W_d\}$ converged under UMT pre-training and back-translation, fine-tune model $\theta$.

---
1: **repeat**
2:    Sample a mini-batch $B$ of either $z \sim \mathcal{D}_s$ or $z \sim \mathcal{D}_t$
3:    Update encoder and its embeddings $\{\theta_e, W_e\}$ with one EBT step starting from $z$ (with Algorithm 2);
4:    Sample a new mini-batch $B$ of either $z \sim \mathcal{D}_s$ or $z \sim \mathcal{D}_t$
5:    Update decoder and its embeddings $\{\theta_d, W_d\}$ with one EBTD step starting from $z$ (with Algorithm 3);
6:    Sample a new mini-batch $B$ of either $z \sim \mathcal{D}_s$ or $z \sim \mathcal{D}_t$
7:    Update full model $\theta$ with a BT step starting from $z$;
8: **until** Convergence

---

glish and Romanian, have distinct bilingual BPE dictionaries. Following BPE, sentences with length 175 or greater are removed. Models are scored on the WMT'14 English-French, WMT'16 English-German and WMT'16 English-Romanian parallel test sets (Koehn et al., 2007). We evaluate the models with BLEU scores (Papineni et al., 2002; Post, 2018) following the same setup of previous works (Song et al., 2019; Nguyen et al., 2021).

**Model Configuration**  We fine-tune the open-source, MASS[6] and CBD[7] UMT model checkpoints provided publicly by Song et al. (2019); Nguyen et al. (2021). These checkpoints have already been fined-tuned with back-translation in large resource settings. We provide separate experiments with the pre-trained XLM[8] model checkpoints provided by Conneau and Lample (2019). These models have undergone generative pre-training but have not yet been trained with back-translation or machine translation objectives. All models are encoder-decoder Transformers with 6 layers and 1024 dimensions. Models are bilingual, respective to their cited language pair, and use the

---

[4]Whenever $x$ and $y$ are of different lengths, we truncate the longer one to have the same length.

[5]https://data.statmt.org/news-crawl/

[6]https://github.com/microsoft/MASS

[7]https://github.com/nxphi47/multiagent_crosstranslate

[8]https://github.com/facebookresearch/XLM

| Method | en - fr | fr - en | en - de | de - en | en - ro | ro - en |
|---|---|---|---|---|---|---|
| NMT (Lample et al., 2018c) | 25.1 | 24.2 | 17.2 | 21.0 | 21.1 | 19.4 |
| PBSMT (Lample et al., 2018c) | 27.8 | 27.2 | 17.7 | 22.6 | 21.3 | 23.0 |
| Multi-Agent (Wang et al., 2019) | - | - | 19.3 | 23.8 | - | - |
| XLM (Conneau and Lample, 2019) | 33.4 | 33.3 | 26.4 | 34.3 | 33.3 | 31.8 |
| MASS* (Song et al., 2019) | 37.5 | 34.8 | 28.3 | 35.2 | 35.0 | 33.0 |
| MASS + QBT-Synced | 37.8 | 35.2 | 28.7 | 35.5 | 35.2 | 33.2 |
| CBD* (Nguyen et al., 2021) | 38.2 | 35.5 | 29.6 | 35.5 | - | - |
| CBD + QBT-Synced | 38.2 | 35.5 | 29.7 | 35.9 | - | - |

Table 1: BLEU scores on the WMT'14 English-French, WMT'16 English-German and WMT'16 English-Romanian unsupervised translation tasks. * marks results obtained with published checkpoints. For QBT-Synced, our model is initialized with the open-source, fine-tuned UMT model MASS (Song et al., 2019) or CBD (Nguyen et al., 2021). Translations are measured with the Moses multi-bleu.perl script (Koehn et al., 2007) as to be comparable with previous work. We also show additional results with SacreBLEU (Post, 2018) in Appendix C for future references.

same corresponding BPE vocabularies.

**Implementation Details** For all configurations the Adam optimizer (Kingma and Ba, 2015) with $\beta_1$ of 0.9 and $\beta_2$ of 0.98 is used for training, with a learning rate of 1e-4. During inference, we use the default greedy decoding for the XLM and MASS model (Conneau and Lample, 2019; Song et al., 2019). For CBD (Nguyen et al., 2021) we use a beam size of 5, and length penalty of 0.6. For all encoder-generated sequences we use the argmax of the output-layer probabilities for NAR decoding. More details can be found in the Appendix B.

## 6 Results and Analysis

### 6.1 Large Scale WMT Experiments with QBT-Synced

We use QBT-Synced to fine-tune the state-of-the-art UMT models available (Song et al., 2019; Nguyen et al., 2021), and test their performance with only 10,000 training steps. We run experiments on an 8-GPU system. The QBT-Synced algorithm 1 is run, using iterative batch updates of EBT, EBTD, and BT steps.

**Main Results** Table 1 shows the performance of our proposed method in comparison with recent UMT methods. For MASS fine-tuning (middle section), the proposed method slightly improves the initial UMT model by an average of 0.3 BLEU points. For CBD fine-tuning (bottom section), the proposed method performs similarly (with +0.1 BLEU points on average). The CBD model is trained via distillation of the XLM and MASS fine-tuned models, and the QBT model may be experiencing diminishing returns on data diversity

| | en - fr | fr - en | en - de | de - en |
|---|---|---|---|---|
| Initialization | 37.5 | 34.8 | 28.3 | 35.2 |
| BT | 37.2 | 33.2 | 26.6 | 34.0 |
| EBT | 30.5 | 28.8 | 19.5 | 24.4 |
| EBTD | 32.05 | 29.5 | 22.4 | 26.0 |
| EBT + EBTD | 28.9 | 24.8 | 20.8 | 26.1 |
| BT + EBTD | 37.0 | 24.1 | 26.5 | 33.9 |
| BT + EBT | 37.7 | 35.0 | 28.4 | 35.4 |
| QBT-Synced | **37.8** | **35.2** | **28.7** | **35.5** |

Table 2: Ablation study of the proposed methods for the MASS fine-tuned model. BLEU scores are presented on the WMT'14 English-French and WMT'16 English-German reference test-sets. QBT-Synced could otherwise be labeled BT + EBT + EBTD.

compared to the MASS experiments. Whereas the CBD model requires two separate, high performance UMT models, our method is able to improve almost all models with no additional parameters and relatively little computation.

**Components of QBT-Synced** In order to study the impact of each individual optimization step in the QBT-Synced procedure, we evaluate the effectiveness of EBT, EBTD, and BT individually and in combination in Table 2. All permutations of BT, EBT, and EBTD are measured after 10,000 training steps. For model initialization, we use the UMT fine-tuned English-French and English-German MASS checkpoints (Song et al., 2019).

We find that the back-translation (BT only) fine-tuned model scores lower on the test set than the initialization. This decline is likely due to the change in training regime from the original, and the lower number of sequences per batch in our setup (1k

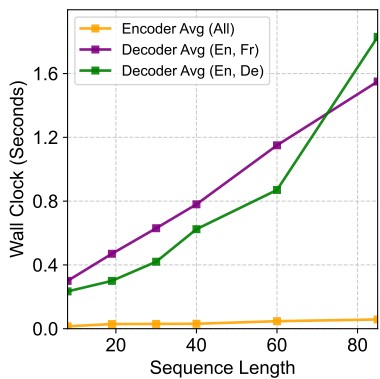
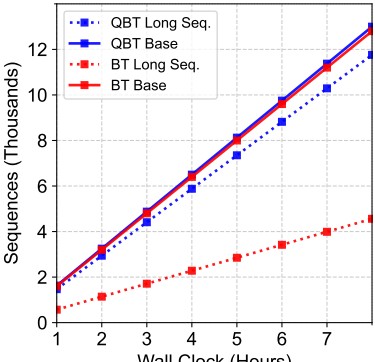
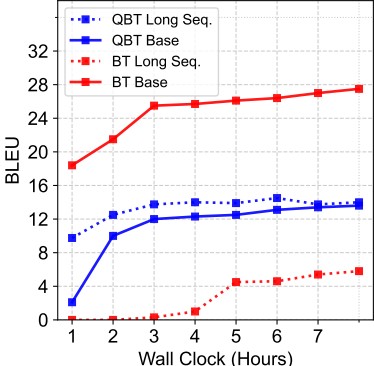

Figure 2: Average forward step run-time.

Figure 3: Data throughput on En and De train sets.

Figure 4: Avg En-De and De-En scores. Here BT is not used during QBT.

versus 2k) (Song et al., 2021). All models underperform the initialization except BT + EBT and QBT-Synced (BT + EBT + EBTD). Without the BT objective all models decline. The model may be losing its ability to back-translate its own high quality outputs, diminishing the final quality. EBT + BT improves the model within 0.1 BLEU of the full QBT-Synced procedure. We speculate that the EBT + BT step serves to better align the encoder and decoder to the final test objective: machine translation.

## 6.2 Limited Resource WMT Experiments with QBT-Staged

We use QBT-Staged to fine-tune a language modeling pre-trained model that has *not* yet undergone back-translation for MT specialization. We limit the computational resource to a total training time of up to 32 hours, with a single RTX 3090 GPU. The XLM pre-trained Transformer is used as our model initialization, which has not yet undergone any MT or BT tuning (Conneau and Lample, 2019). The baseline is fine-tuned with the BT and DAE objectives. The QBT-Staged encoder is warmed up with randomly paired monolingual samples from different language corpora for 5,000 training steps, as described in section 4.3. The remaining training budget is split as follows: 4 hours of EBT, 16 hours of EBTD, and 12 hours of BT.

**Main Results** Results are shown in Table 3. QBT-Staged model outperforms the baseline on all language directions, excluding French to English in which they tie. Our limited resource QBT-Staged model even outperforms the original 8-GPU implementation from Conneau and Lample (2019) on

|  | en - fr | fr - en | en - de | de - en |
|---|---|---|---|---|
| BT Baseline | 32.9 | 32.0 | 25.2 | 31.2 |
| QBT-Staged | 34.9 | 32.0 | 25.9 | 31.9 |

Table 3: BLEU scores on the *limited resource* WMT'14 English-French and WMT'16 English-German UMT tasks. The XLM pre-trained model is used as our initialization.

English to French by 1.5 BLEU points, compared to their score of 33.4 for UMT achieved from XLM initialization.

**QBT Efficiency** To demonstrate the speed gains of using encoder for NAR generation during QBT procedures, Figure 2 presents average wall clock time of translation generation on batch-size 32 samples from the WMT test set. We report the average wall clock time per batch, which is sampled to be of approximately uniform sequence length. For the encoder, we average wall-clock time for all translation directions, as the difference is not visually distinguishable. For the decoder, language directions are presented independently. We can see the drastically reduced run-time of the generation when using the repurposed NAR encoder compared with the AR decoder, which greatly saves time and energy for UMT training with QBT.

## 6.3 Long Sequence Translation

The WMT dataset is skewed towards short sequences, with the majority of sequence shorter than 50 token length. Therefore the large resource main results in Table 1 and limited resource results in Table 3 presented do not portray the dramatic runtime benefits of encoder back-translation versus

decoder back-translation. In this section, we filter the German and English WMT train datasets to contain only sequences longer than length 120. For evaluation, we use the full WMT'16 German-English test set with no minimum sequence length, as there are not enough long sequences.

We run 4 independent experiments: baseline BT on unfiltered data, QBT-Staged on unfiltered data, BT on long-sequence data, and QBT-Staged on long-sequence data. All models are allocated 8 hours of training time. QBT-Staged involves 0.5 hours of EBT, and uses EBTD for the remaining time.[9]

**Data Throughput** In Figure 3, QBT algorithm demonstrates almost negligible run-time slowdown at longer sequence length during training. However, the BT algorithm sees only about one third of the data during training. QBT and BT data throughput are similar at base sequence length.

**Training Curves** In Figure 4, the BT algorithm considerably under-performs the others on long sequence length. Surprisingly, the QBT model trained on only long sequences outperforms the baseline QBT model. We can also see the QBT models converge faster due to larger data throughput. The encoder generates sequences in a token-to-token fashion, and it is possible that this translation configuration generalizes well to sequences of unseen length. Additionally, the long-sequence encoder has seen more tokens during training, and effectively more training data. Though we only score decoder outputs, the QBT decoder sits on top of the EBT-trained encoder representation, and may be benefiting from these advantages.

### 6.4 Unsupervised Programming Language Translation

In order to more thoroughly test our proposed method, we provide a demo of QBT on unsupervised programming language translation.

For dataset curation we follow the procedure from Roziere et al. (2020). Monolingual Python and Java source code is downloaded using Google's BigQuery.[10] Source code is pre-processed using the Python6 tokenizer[11] and javalang5 tokenizer[12]

| Model | Beam Size | Python-Java | Java-Python |
|-------|-----------|-------------|-------------|
| BT | 1 | 25.61 | 22.57 |
| QBT-Staged | 1 | 33.75 | 32.27 |
| BT | 10 (top 1) | 38.26 | 37.86 |
| QBT-Staged | 10 (top 1) | 41.49 | 40.89 |

Table 4: BLEU scores on the *filtered* GeeksForGeeks parallel test set.

respectively. A BPE vocabulary of size 30k is learned over the union of tokenized Python and Java datasets. Data is filtered to include only static functions and code comments are removed. We filter out sequences longer than length 100 as outliers. In total we keep 10 million code snippets of Python and Java each. For evaluation, we use the GeeksforGeeks parallel test set provided by Roziere et al. (2020).[13] A Transformer model with 6 encoder layers and 4 decoder layers is used. The baseline is fine-tuned with BT for 24 hours. For QBT, we use QBT-Staged and train with EBT, EBTD, then BT, for 8 hours each. Final models are evaluated with greedy decoding and with beam 10 (select top 1) search (Roziere et al., 2020).

**Results** QBT outperforms the BT baseline for both programming language directions and decoding setups. The performance gap between the models decreases at a higher beam-size. This shift may be due to the encoder-synthesis more strongly resembling greedy decoder generations.

### 6.5 Discussion

**Translation Examples** The qualitative differences in model translations following BT or QBT is of great interest. In Table 5 we provide samples from the fine-tuned German to English limited resource translation task in section 6.2. Here the EBT (encoder generation) and QBT (decoder generation) models provide a remarkably similar translation. The BT model suffers some syntactic degeneration, likely due to the fact that the BT model may have not fully converged given the limited time budget.

**Self-BLEU Analysis** The translation comparison raises the question: does QBT work to align the language representations between the encoder and decoder? The Self-BLEU metric, originally proposed by Zhu et al. (2018), uses a model-generated corpus

---

[9]Note that here BT is not involved to separately demonstrate the efficiency of QBT on top of BT.

[10]https://cloud.google.com/bigquery

[11]https://github.com/c2nes/javalang

[12]https://docs.python.org/3/library/tokenize.html

[13]Due to long sequences filtered out, our results compare BT and QBT in a novel scenario, but should not be compared to the TransCoder model from Roziere et al. (2020).

| Model | Generation |
|-------|-----------|
| Target | It is a mixture of a huge fashion show and a waxworks chamber of horrors: designer Jean Paul Gaultier (63) presented an exhibition of is work in Munich on Wednesday. |
| EBT | It's a mix of very big fashion show and a Wachsfiguren-gruselkabinett: French designer Jean Paul Gaultier (63) opened on Wednesday in Munich a exhibition celebrating his work collection. |
| QBT-Staged | It is a combination of a big Modenschau and a Wachsfiguren grusele-skater : French designer Jean Paul Gaultier ( 63 ) opened on Wednesday in Munich a show about his work career. |
| BT | It's a combination of a complete Modenschau and a Wachsfiguren-looking Gruselkabinett Gruselkabinett: The designer Jean Paul Gaultier (63), who has a collection of his own pieces, including a collection of his own pieces, including a collection of his own pieces, including a collection of his own pieces, a collection of his own pieces, including a collection |

Table 5: Comparison of translations following QBT-staged training in 6.2. From the WMT'14 En-De test set.

as the BLEU reference text, acting as a similarity metric between any two corpora. In the following, we score the outputs of the converged EBT encoder against various references. For the XLM model, we use the converged QBT-Staged checkpoints from Table 3, and for MASS we use the QBT-Sync ablation checkpoints in Table 2. Scores are shown in Figure 5. Decoder BT denotes the encoder translations scored against a separate baseline BT decoder translations. Decoder EBT uses decoder translations following EBT training of the model encoder. Finally, Decoder EBTD depicts the decoder translations following EBT and EBTD.[14] Across all four scenarios, EBTD increases the Self-BLEU of the decoder to the encoder translations. This relationship suggests that there is potentially knowledge distillation occurring during EBTD. Note as well that the XLM models presented, trained with QBT-Staged, underwent more steps of EBT and EBTD than the MASS models. This is reflected in the higher overall self-BLEU scores of XLM versus MASS.

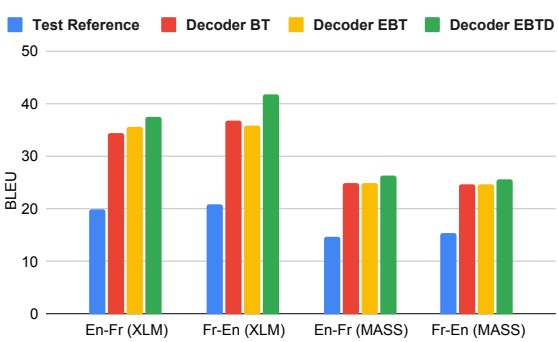

Figure 5: BLEU and self-BLEU scores for encoder-generated sequences against various references. The standard BLEU scores for encoder translations are given in Test Reference, Decoder BT is the self-BLEU of the encoder vs the baseline BT decoder, and so on. All source sequences are from the WMT'14 English-French test set.

## 7 Conclusion

We proposed Quick Back-Translation (QBT) for unsupervised machine translation, which endows translation capabilities to the Transformer encoder, and leverages encoder-generated sequences for improved performance and training speed for UMT. QBT combines Back-Translation (BT), Encoder Back-Translation (EBT), and Encoder Back-Translated Distillation (EBTD). This optimization framework creates efficiency gains, and provides new synthetic data sources. QBT achieved similar fine-tuned performance to other state-of-the-art UMT models with no additional parameters and higher data efficiency. On long sequences, QBT considerably outperforms the original back-translation algorithm in data throughput and translation quality.

---

[14]For MASS Decoder EBT is the EBT + BT model from the ablation in Table 2. This model is used as it avoids the decoder collapse in the QBT-Sync setup. Decoder EBTD uses the final QBT-Staged model from Table 3, and the final QBT-Sync model as in our main results Table 1, for XLM and MASS respectively.

## 8  Limitations

Our QBT methods serve as an augmentation of the standard BT algorithm for learning unsupervised machine translation models, mainly with efficiency considerations in low computational resource scenarios. They still can not fully replace the standard BT procedures, as QBT still needs BT inside the algorithms to achieve competitive translation qualities. Even so, our performance gain compared to previous state-of-the-art UMT models are modest, whereas we mainly gain in training speed. Due to the limited resource setups, the models that we experimented are considered relatively small in sizes compared with the large language models nowadays. It would be interesting to explore further on the larger scales. Our methods currently apply to encoder-decoder Transformer architectures, which may limit the applications. For different model architectures, we may hope to apply similar ideas of using a fast (and potentially separate) generation model to generate synthetic data for BT-like algorithms. Furthermore, the use of encoder for back-translation (EBT) with NAR generation is also limited in that the output sequence length can not go beyond the input sequence length, which we hope to relax with more sophisticated NAR strategies in the future.

## Acknowledgments

We express our gratitude to Dr. Nakul Verma for his support and insight over the course of this work.

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

# A QBT Algorithm Descriptions

We detail the algorithmic procedures for QBT components: batch EBT for encoder translation tuning in Algorithm 2, batch EBTD for decoder generation tuning in Algorithm 3. We also describe the complete procedure for QBT-Staged in Algorithm 4.

---

**Algorithm 2** Encoder Back-Translation (EBT) Step: Given two monolingual corpora $\mathcal{D}_s, \mathcal{D}_t$ and a Transformer model $\theta = \{\theta_e, W_e, \theta_d, W_d\}$, train a UMT model over the encoder $\{\theta_e, W_e\}$.

---

1: Given a mini-batch $B$ sampled with either $z \sim \mathcal{D}_s$ or $z \sim \mathcal{D}_t$
2: **for** $z \in B$ **do**
3:    Infer translations $z'$ with encoder $\{\theta_e, W_e\}$;
4:    Predict translations $z''$ conditioned on $z'$ with the same encoder;
5:    Update encoder parameters $\{\theta_e, W_e\}$ with gradient computed from reconstruction loss between target $z$ and prediction distribution for $z''$;
6: **end for**

---

**Algorithm 3** Encoder Back-Translated Distillation (EBTD) step: Given two monolingual corpora $\mathcal{D}_s$, $\mathcal{D}_t$ and a Transformer model $\theta = \{\theta_e, W_e, \theta_d, W_d\}$, train a UMT model over the $\theta$.

---

1: Given a mini-batch $B$ sampled with either $z \sim \mathcal{D}_s$ or $z \sim \mathcal{D}_t$
2: Freeze all encoder parameters $\{\theta_e, W_e\}$
3: **for** $z \in B$ **do**
4:    Infer translations $z'$ with encoder $\{\theta_e, W_e\}$;
5:    Predict translations $z''$ conditioned on $z'$ with the full encoder-decoder Transformer $\theta$;
6:    Update only the decoder parameters $\{\theta_d, W_d\}$ with gradients computed from reconstruction loss between target $z$ and prediction distribution for $z''$;
7: **end for**

---

**Algorithm 4** Quick Back-Translation Staged (QBT-Staged): Given two monolingual corpora $\mathcal{D}_s, \mathcal{D}_t$, and an encoder-decoder Transformer model $\theta = \{\theta_e, W_e, \theta_d, W_d\}$ undergone general pre-training without MT capability, fine-tune model $\theta$ for UMT.

---

1: **Stage 1:** encoder warmup with random translation
2: **repeat**
3:    Sample a mini-batch $B$ composed of random pairs $(x, y)$, with $x \sim \mathcal{D}_s, y \sim \mathcal{D}_t$, independently;
4:    Truncate $x$ or $y$ to make the pairs have equal length;
5:    Predict translation $y'$ conditioned on $x$ only using the encoder $\{\theta_e, W_e\}$ with computation described in Equation (2);
6:    Update the encoder parameters $\{\theta_e, W_e\}$ with gradient computed from reconstruction loss between $y'$ and target $y$;
7: **until** Convergence of a fixed amount of steps/time
8: **Stage 2:** EBT to train the encoder for translation
9: **repeat**
10:    Sample a mini-batch $B$ of either $z \sim \mathcal{D}_s$ or $z \sim \mathcal{D}_t$
11:    Update encoder and its embeddings $\{\theta_e, W_e\}$ with EBT starting from $z$ (with Algorithm 2);
12: **until** Convergence of a fixed amount of steps/time
13: **Stage 3:** EBTD to train the decoder for generation
14: **repeat**
15:    Sample a mini-batch $B$ of either $z \sim \mathcal{D}_s$ or $z \sim \mathcal{D}_t$
16:    Update decoder and its embeddings $\{\theta_d, W_d\}$ with EBTD starting from $z$ (with Algorithm 3);
17: **until** Convergence of a fixed amount of steps/time
18: **Stage 4:** BT to tune the full UMT model
19: **repeat**
20:    Sample a mini-batch $B$ of either $z \sim \mathcal{D}_s$ or $z \sim \mathcal{D}_t$
21:    Update full model $\theta$ with BT starting from $z$;
22: **until** Convergence of a fixed amount of steps/time

---

|       | MASS* (Song et al., 2019) | MASS + QBT-Synced |
|-------|-----|-----|
| en - fr | 37.5 | 38.1 |
| fr - en | 34.8 | 35.3 |
| en - de | 28.4 | 29.0 |
| de - en | 35.4 | 36.0 |
| en - ro | 35.1 | 35.5 |
| ro - en | 33.1 | 33.2 |

Table 6: BLEU scores on WMT'14 En-Fr, WMT'16 En-De, WMT'16 En-Ro. MASS* are measured from published UMT fine-tuned checkpoints from the MASS work (Song et al., 2019). Translations are scored with the SacreBLEU scripts (Post, 2018).

## B   Implementation Details

All methods are implemented following the original codebases of relevant multilingual pre-trained models. In our main UMT results in Section 6.1 a batch size of 16 and 1k maximum tokens per batch are used. Our limited resource scenario in Section 6.2 uses a batch size of 32 and 2k maximum tokens per batch. For all configurations the Adam optimizer (Kingma and Ba, 2015) with $\beta_1$ of 0.9 and $\beta_2$ of 0.98 is used for training, with a learning rate of 1e-4. Best model checkpoint is selected by validation BLEU scores from greedy decoding. During inference, we use the default greedy decoding for the XLM and MASS model (Conneau and Lample, 2019; Song et al., 2019). For CBD (Nguyen et al., 2021) we use a beam size of 5, temperate of 1, and length penalty of 0.6. For all encoder-generated sequences we use the argmax of the output-layer probabilities for NAR decoding. During Encoder Back-Translation (EBT), we occasionally observe mode-collapse, in which the model copies the input sequence as the output sequence. In this case, we penalize sequence copying by adding as a regularization term the inverse of the negative log-likelihood against the source sentence. The penalization is applied at each EBT step.

## C   Complementary Results: SacreBLEU

Our main results in Table 1 use the same Moses Perl BLEU evaluation (Koehn et al., 2007) and byte pair encoding vocabularies as previous UMT work (Conneau and Lample, 2019; Song et al., 2019; Nguyen et al., 2021). This choice of Moses BLEU script is to follow previous work for fair compari-

son. In order to further validate our methods, we conduct experiments separately using the Scare-BLEU metric implementation (Post, 2018). This metric allows for a more controllable metric comparisons for future works.

In Table 6 our models are trained with QBT-Sync, following the exact same procedure as our Moses BLEU results. Notably here we differ in that the best checkpoint is selected via the highest validation on the SacreBLEU metric. Scores in SacreBLEU are reported. Similar to before, we find that QBT-Sync has a distinct but slight performance bump over the MASS baseline.