# OpenReview forum: "Quick Back-Translation for Unsupervised Machine Translation"
_EMNLP/2023/Conference — EMNLP 2023 Findings_

### Official Review · Reviewer_zNyd · 2023-08-01

**Soundness:** 3

**Excitement:**

3: Ambivalent: It has merits (e.g., it reports state-of-the-art results, the idea is nice), but there are key weaknesses (e.g., it describes incremental work), and it can significantly benefit from another round of revision. However, I won't object to accepting it if my co-reviewers champion it.

**Missing References:**

N/A

**Paper Topic And Main Contributions:**

The passage discusses recent advancements in unsupervised machine translation, specifically focusing on the marriage of the Transformer model and the back-translation algorithm. While the Transformer and back-translation are powerful techniques, they have certain limitations that the authors aim to address with their proposed method called Quick Back-Translation (QBT). QBT repurposes the Transformer's encoder as a generative model, utilizing encoder-generated sequences to train the decoder in conjunction with the traditional autoregressive back-translation. This approach improves data throughput and utilization, addressing the runtime issue of autoregressive inference during back-translation and the lack of synthetic data efficiency.

The article is well written. The main problem is that the proposed method has limited improvement in the translation quality of unsupervised MT, and the evaluation metric is relatively single.


**Questions For The Authors:**

Q1: For Table 2, why is EBTD better than EBT, but BT+EBTD is worse than BT+EBT? And BT+EBTD is generally lower than BT?

Q2: For Table 3, why is the improvement (2BLEU) on En-Fr significantly larger than the improvement on the rest of the translation directions?


**Reasons To Accept:**

-	The introduction of NAR generation in QBT effectively improves the translation efficiency.
-	The article is well written and easy to follow.


**Reasons To Reject:**

-	QBT just changes the AR structure in BT to NAR structure, and its novelty is limited.
-	Translation quality improvements are limited. The main experimental results in Table 1 show that QBT can only bring about an improvement of about 0.2 BLEU, which I think is almost no improvement. For such a subtle improvement, the authors should adopt more metrics beyond BLEU (such as COMET), as well as a significance test to prove the effectiveness of the proposed method.
-	The efficiency evaluation in Figure 2 is less standard. As far as I know, the speedup ratio in NAT is often calculated at batch-size=1, and it will give additional efficiency improvements under different batch-sizes.


**Reproducibility:**

4: Could mostly reproduce the results, but there may be some variation because of sample variance or minor variations in their interpretation of the protocol or method.

**Reviewer Confidence:**

4: Quite sure. I tried to check the important points carefully. It's unlikely, though conceivable, that I missed something that should affect my ratings.

**Typos Grammar Style And Presentation Improvements:**

-	Paper is well written.

---

> ### Author Rebuttal · Authors · 2023-08-29
>
> Thank you very much for the constructive review and comments! We provide some responses below based on your valuable insights.
>
> 1. "The introduction of NAR generation in QBT effectively improves the translation efficiency."
>
> While we understand the limited novelty of BT for a NAR model in general, we would like to emphasize the unique demonstration of NAR BT inside an existing AR model structure, without any extra parameters or changes to the AR capability or quality. In particular, QBT demonstrates that existing AR MT models contain a highly efficient NAR model within their architecture, which allows for efficient model training (especially at long sequence lengths) and synthetic data diversification. Previous BT and NAR methods require dramatic modifications to the Transformer encoder-decoder, and suffer in quality from the loss of AR capability [1][2].
>
> [1] Junliang Guo, Xu Tan, Linli Xu, Tao Qin, Enhong Chen, and Tie-Yan Liu. 2020. Fine-tuning by curriculum learning for non-autoregressive neural machine translation. In Proceedings of the AAAI Conference on Artificial Intelligence, volume 34, pages 7839–7846\
> [2] Zhenqiao Song, Hao Zhou, Lihua Qian, Jingjing Xu, Shanbo Cheng, Mingxuan Wang, and Lei Li. 2021. switch-glat: Multilingual parallel machine translation via code-switch decoder. In International Conference on Learning Representations.
>
> 2. "Translation quality improvements are limited. The main experimental results in Table 1 show that QBT can only bring about an improvement of about 0.2 BLEU, which I think is almost no improvement. For such a subtle improvement, the authors should adopt more metrics beyond BLEU (such as COMET), as well as a significance test to prove the effectiveness of the proposed method."
>
> It is a good point that more metrics, such as COMET, along with significance tests, would provide more comprehensive comparisons of the model performances. We will make these measurements to bolster the BLEU scores. Nevertheless, we are in agreement that the translation quality improvements are only modest in the large-scale WMT scenario, and we will make this clear in our paper revision. Please note that the most comparable method to ours in terms of scores and methodology, Cross-Model Back-Translated Distillation [1], requires three, large, pre-trained models. Two of these models must be independently, extensively fine-tuned with BT. In total, 3x the number of parameters are required. Whereas QBT improves BLEU without any additional model parameters or auxiliary models. The benefits of our method are more in the efficiency gain with similar translation qualities compared to previous best models. For example, in resource constrained and long sequence scenarios, the gained efficiency enables very strong BLEU scores versus the original BT algorithm.
>
> [1] Nguyen, X.P., Joty, S., Nguyen, T.T., Wu, K. and Aw, A.T., 2021, July. Cross-model back-translated distillation for unsupervised machine translation. In International Conference on Machine Learning (pp. 8073-8083). PMLR.
>
> 3. "The efficiency evaluation in Figure 2 is less standard. As far as I know, the speedup ratio in NAT is often calculated at batch-size=1, and it will give additional efficiency improvements under different batch-sizes."
>
> Thank you for highlighting this. In conducting this experiment, we deliberated between using batch size 1 for Figure 2, versus the batch size 32 used in our full experiment (Figures 3 and 4). We opted to use the batch size from the training scenario. But we agree that using batch size 1 would more directly probe the inference runtime for these models.
>
> Below we have calculated the results with batch size 1. Sequences are taken from the WMT test sets, as before. As there are only 3000 samples in the test set, we bucket the source sequences based on their lengths inside a window for detailed analysis. Sequences with length +/-5 of the value in the *Sequence Length* column are taken, a random sample of 5 of these are passed through the corresponding model, and the average of the run-times is taken. (En, Fr) is the average of the En-Fr and Fr-En direction, and so on for the pairs. For encoder generations, output sequence lengths are the same as source sequence lengths, thus the generation runtime for all language directions are similar so we take the average of different language pairs. Decoder means the full AR model, and we present the runtime for the two language pairs separately.
>
> Overall, we observe that the encoder is considerably faster for the encoder than the AR mode. At a batch size of 1, the encoder runtime is not as sensitive to the increasing sequence length in our sample. Meanwhile the decoder generation experiences a slowdown of up to half a second, and over 1 second / sample in the batch size 32 case.
>
> We will also add this information to the final paper.
>
> | Batch Size = 1: Forward step runtime (seconds), averages |                      |                  |                  |
> | -------------------------------------------------------- | -------------------- | ---------------- | ---------------- |
> |       Sequence Length                        | Encoder (En, Fr, De) | Decoder (En, Fr) | Decoder (En, De) |
> | 8                                                        | 0.009                | 0.24             | 0.19             |
> | 19                                                       | 0.009                | 0.28             | 0.22             |
> | 30                                                       | 0.01                 | 0.4              | 0.36             |
> | 40                                                       | 0.012                | 0.45             | 0.41             |
> | 60                                                       | 0.01                 | 0.57             | 0.48             |
> | 85                                                       | 0.011                | 0.85             | 0.71             |
> |                                                          |                      |                  |                  |
>
> |Batch size 32: Forward step runtime (seconds), averages  |                      |                  |                  |
> | -------------------------------------------------------- | -------------------- | ---------------- | ---------------- |
> |                                                          | Encoder (En, Fr, De) | Decoder (En, Fr) | Decoder (En, De) |
> | 8                                                        | 0.015                | 0.3              | 0.234            |
> | 19                                                       | 0.029                | 0.47             | 0.3              |
> | 30                                                       | 0.03                 | 0.63             | 0.42             |
> | 40                                                       | 0.031                | 0.78             | 0.624            |
> | 60                                                       | 0.047                | 1.15             | 0.87             |
> | 85                                                       | 0.058                | 1.55             | 1.83             |
>
> 4. "Q1: For Table 2, why is EBTD better than EBT, but BT+EBTD is worse than BT+EBT? And BT+EBTD is generally lower than BT?"
>
> This is a great clarifying question. For EBT and EBTD in this ablation, the decoder experiences a severe degradation in translation quality, as we are changing the model weights with disregard to the already converged MT capability. This highlights how for QBT-Sync, it is important to use BT during EBT and EBTD, to at least maintain the decoder capabilities. In the EBT row, we are shifting the encoder representations away from what the frozen decoder is expecting, which seems to be detrimental to the decoder translations. Whereas in the EBTD row, the encoder stays frozen, and the decoder is made to back-translate very low-quality, noisy translations back to the original sequence. In this way, EBTD acts as a very lossy de-noising objective, whereas EBT completely corrupts the encoder-decoder model.
>
> Once we incorporate BT, the decoder can adapt to the shifting encoder representation on each step. So EBT + BT  aligns the encoder with the translation objective, while maintaining the BT capabilities of the decoder. For EBTD + BT, note that the encoder has never been trained for MT. So EBTD + BT is similar to BT with an additional very-lossy noising objective, which we find does not improve over BT. We hope that this clarifies the roles of the optimizations encompassing QBT-Sync. We will add additional exposition to our paper on this point.
>
> 5. "Q2: For Table 3, why is the improvement (2BLEU) on En-Fr significantly larger than the improvement on the rest of the translation directions?"
>
> The high score on En-Fr was a bit surprising. Our score of 34.9 was higher than the original XLM score of 33.4 [1], which used 8-GPUs vs our single GPU (although both are lower than fine-tuned MASS). While the exact reason for this dramatic difference in En-Fr over other language directions needs further investigation, we conjecture that word-level statistics of different pairings of languages benefit the bidirectional encoder in certain scenarios, especially where translations are more “simple”, vs translations which require widely varying sequence lengths between source and target languages. We will investigate this more to gain a better understanding.
>
> [1] Conneau, A. and Lample, G., 2019. Cross-lingual language model pretraining. Advances in neural information processing systems, 32.

---

### Official Review · Reviewer_w5DB · 2023-08-04

**Soundness:** 4

**Excitement:**

4: Strong: This paper deepens the understanding of some phenomenon or lowers the barriers to an existing research direction.

**Paper Topic And Main Contributions:**

The reviewed paper focuses on unsupervised machine translation by introducing Quick Back-Translation (QBT). The paper outlines the existing challenges in unsupervised machine translation, particularly the computational load of autoregressive inference during back-translation and the synthetic data inefficiency of the process. The authors repurpose the Transformer encoder, harnessing it as a generative model. By utilizing encoder-generated sequences to refine the decoder, QBT significantly augments data throughput and utilization.
The experimental evaluation conducted on various WMT benchmarks serves as a testament to the effectiveness of QBT. QBT outperforms the standard back-translation method in terms of training efficiency while maintaining comparable translation quality.

**Questions For The Authors:**

- In the experimental evaluation, could you discuss any specific instances or examples where QBT demonstrated a notable improvement over the standard approach in terms of translation quality?

- Could you provide insights into the mechanism through which QBT enhances data efficiency? What is lost in the absence of the autoregressive decoder? did you encounter any trade-offs or potential drawbacks in other aspects of the model's performance, such as encoding quality or training stability?

- Are there any specific characteristics of the Transformer architecture that make it particularly amenable to being repurposed for generative purposes, as demonstrated in QBT?





**Reasons To Accept:**

- The clarity with which the paper presents its proposed methodology, supported by experiments and outcomes
- Novel approach for the field of unsupervised machine translation

**Reasons To Reject:**

- further analysis on the specific mechanisms through which the encoder-generated sequences contribute to decoder improvement could provide a deeper understanding of QBT.

**Reproducibility:**

4: Could mostly reproduce the results, but there may be some variation because of sample variance or minor variations in their interpretation of the protocol or method.

**Reviewer Confidence:**

3: Pretty sure, but there's a chance I missed something. Although I have a good feel for this area in general, I did not carefully check the paper's details, e.g., the math, experimental design, or novelty.

---

> ### Author Rebuttal · Authors · 2023-08-29
>
> Thank you very much for the careful review and constructive comments! We provide some more details below.
>
> 1. "further analysis on the specific mechanisms through which the encoder-generated sequences contribute to decoder improvement could provide a deeper understanding of QBT."
>
> More rigorously evaluating the encoder synthetic data diversity would be of great value, especially in relation to the decoder-generated translations. We made some initial progress towards this analysis with our Self-BLEU results in Appendix section C. Here we train a model with no UMT pre-training with EBT. We generate translations with this encoder, then use these translations as the BLEU reference for a number of decoder-generation models. Interestingly, models trained with AR BT have Self-BLEU similar to their BLEU scores against the test set. This score implies that the encoder-generated sequences are different, but not that different from the decoder. Another key detail of this analysis is that encoder-distillations, namely EBTD, increase the Self-BLEU between encoder generated outputs and decoder generated outputs.
>
> 2. "In the experimental evaluation, could you discuss any specific instances or examples where QBT demonstrated a notable improvement over the standard approach in terms of translation quality?"
>
> QBT outperforms BT in translation quality in a variety of scenarios. However, the amount by which it outperforms varies, due to the compute resources involved and the characteristics of the dataset (sequence lengths, similarity between source and target languages, and so on). One scenario where QBT shines in translation quality is long sequence translation (Figure 4). In our experiment QBT nearly triples the BLEU score of the original BT method (although both are not at the level of BLEU for short sequences). This outperformance is due in part to the data throughput efficiency of the encoder, but also likely due to properties of bidirectional translation itself.
>
> Below we provide samples from the final fine-tuned German to English translation task in Section 6.2 Limited Resource WMT Experiments. Here the EBT (encoder generation) and QBT (decoder generation) models provide a similar translation. The sequences seem more semantically aligned with the target than the BT output. In addition, the BT model suffers some syntactic degeneration, which is likely due to the fact that the BT model may have not fully converged given the limited time budget.
>
> **Target**: *It is a mixture of a huge fashion show and a waxworks chamber of horrors : designer Jean Paul Gaultier ( 63 ) presented an exhibition of is work in Munich on Wednesday .*
>
> **EBT**: *It 's a mix of very big fashion show and a Wachsfiguren-gruselkabinett : French designer Jean Paul Gaultier ( 63 ) opened on Wednesday in Munich a exhibition celebrating his work collection .*
>
> **QBT (No BT)**: *It is a combination of a big Modenschau and a Wachsfiguren grusele-skater : French designer Jean Paul Gaultier ( 63 ) opened on Wednesday in Munich a show about his work career .*
>
> **BT**: *It 's a combination of a complete Modenschau and a Wachsfiguren-looking Gruselkabinett Gruselkabinett : The designer Jean Paul Gaultier ( 63 ) , who has a collection of his own pieces , including a collection of his own pieces , including a collection of his own pieces , including a collection of his own pieces , a collection of his own pieces , including a collection*
>
> 3. "Could you provide insights into the mechanism through which QBT enhances data efficiency? What is lost in the absence of the autoregressive decoder? did you encounter any trade-offs or potential drawbacks in other aspects of the model's performance, such as encoding quality or training stability?"
>
> Thank you for these great clarifying questions. QBT enhances data efficiency by generating a large quantity of synthetic translations very quickly using the encoder as a generative model. The encoder-as-generative-model is fast to initialize, to train with BT, and scales with time-complexity linear in sequence lengths, versus the quadratic time complexity of the decoder. However, given the encoder’s fixed output length, we find the BLEU maxes out at around 20 for the models tested; the encoder is a weak translator compared to the full AR counterpart. It would be interesting to further explore enhancements to the encoder with NAR sampling techniques [1].
>
> In addition, Encoder Back-Translation is susceptible to “translation collapse”, whereby it simply copies input sequences as its output. For example, given some French, it would output the same French. Sequence copying is a global minimum of the back-translation loss function: if it maps French to French, then when it back-translates to the original French, copying again will result in a loss of 0. To keep the model away from converging into the global minimum, it is important to coerce it into cross-lingual behavior. We accomplish this through a very brief warm-up period where we train the model to map from random French to random English sequences, and vice versa. This results in a momentary degradation of model quality, but aligns the outputs towards the modes of the target language. For example, common words like “the” in English and “le” will appear in corresponding frequencies in the random inputs and targets. As the encoder has a powerful pre-trained representation, it will bias its translation of “the” towards “le”. This technique is interestingly similar to dictionary learning in the statistical machine translation field (Ravi and 035 Knight, 2011a; Klementiev et al., 2012).
>
> After very brief random-MT initialization, EBT typically converges towards the desired local minima. However, in some scenarios the model can collapse to the copying strategy later during EBT training. In order to counteract this, we apply a small regularization term to the loss which penalizes copying (Appendix B). This is calculated as the negative log-likelihood of the model output distribution against the source sequence to discourage direct copying.
>
> [1] Sun, Z., Li, Z., Wang, H., He, D., Lin, Z. and Deng, Z., 2019. Fast structured decoding for sequence models. Advances in Neural Information Processing Systems, 32.
>
> 4. "Are there any specific characteristics of the Transformer architecture that make it particularly amenable to being repurposed for generative purposes, as demonstrated in QBT?"
>
> Yes, absolutely. The QBT method is reliant on the encoder-decoder Transformer architecture. The AR nature of the decoder is largely due to the casual/unidirectional mask of the self-attention, and most of the NAR decoder remove this limitation by adopting the bidirectional mask in the decoder. The encoder is naturally equipped with the bidirectional self-attention mask, which blurs its boundaries with an NAR decoder. Without the complexity of utilizing a standalone fully dedicated NAR decoder on top of the Transformer encoder-decoder architecture, repurposing the encoder as an NAR decoder turns out to be a natural choice. All of these are made possible by the flexibility of the Transformer structure with attention mechanisms and parallel processing of sequential tokens by tweaking attention masks.

---

### Official Review · Reviewer_7rp5 · 2023-08-05

**Soundness:** 4

**Excitement:**

4: Strong: This paper deepens the understanding of some phenomenon or lowers the barriers to an existing research direction.

**Paper Topic And Main Contributions:**

The paper introduces a new method to unsupervised machine translation: quick back translation (QBT). QBT is the result of NAR translation from the encoder. It seems to have worse quality (per experiment results), but, when used in the unsupervised MT training procedure (eg QBT synced or QBT staged), can improve the training efficiency (because of the quick translation?) and provides comparable results to the state of the art, or slightly better in some cases/tests. Paper conducted experiments to examine the QBT impact in general, or in special cases (eg, long sequence), showing the positive effect of QBT.

Overall, I like the idea of using the encoder to produce back translation and like to confirm its results compared to BT. The efficiency advantage of QBT probably is more interesting than the quality improvement (which is slight based on the result tables.)

Strength:
- Using encoder to produce BT is interesting and on the novel side.
- Paper proposing several ways of integrating QBT into the training procedure also seems interesting to me.
- Good amount of experiments to examine the effectiveness of QBT.

Weaknesses:
- Strength of result: it seems that translation quality of the new procedure is mostly the same as state of the art.


**Reasons To Accept:**

Strengths above: interesting idea.

**Reasons To Reject:**

The quality improvement seems to be mostly comparable to state of the art.

**Reproducibility:**

4: Could mostly reproduce the results, but there may be some variation because of sample variance or minor variations in their interpretation of the protocol or method.

**Reviewer Confidence:**

4: Quite sure. I tried to check the important points carefully. It's unlikely, though conceivable, that I missed something that should affect my ratings.

---

> ### Author Rebuttal · Authors · 2023-08-29
>
> Thank you for your comprehensive review of our paper. We appreciate your recognition of the novelty in using the encoder for back translation and the various integration approaches we proposed for QBT in the training process. Your observation about the efficiency advantage aligns with our intention to enhance training procedures, making them more feasible even in resource-constrained settings.
>
> We acknowledge your point regarding the comparable quality improvement to the state of the art. We believe that QBT's ability to efficiently approach these moderate gains, along with its advantages in scenarios with limited resources and longer sequences, emphasize its practicality and utility for UMT.
>
> We have provided some new details in our other two responses.

---

### Meta-Review · Area_Chair_hxLW · 2023-09-17

**Recommendation:** 4

**Metareview:**

This paper introduces a method (quick back-translation) that can mostly be seen as an efficiency improvement for unsupervised MT, re-using the encoder of a unsupervised MT training setup to perform non-autoregressive decoding instead of using a separate, autoregressive model.

strengths:
- on average, reviewers consider the novelty one of the strengths of the paper
- writing and extensive experimentation are convincing.

weaknesses:
- improvements in quality are minimal. The main benefit is training efficiency for unsupervised MT, which has lower significance than demonstrating quality improvements, or applicability beyond unsupervised MT.

---

### Decision · Program_Chairs · 2023-10-07

**Decision:**

Accept-Findings

**Comment:**

This paper introduces a method (quick back-translation) that can mostly be seen as an efficiency improvement for unsupervised MT, re-using the encoder of a unsupervised MT training setup to perform non-autoregressive decoding instead of using a separate, autoregressive model.

strengths:
- on average, reviewers consider the novelty one of the strengths of the paper
- writing and extensive experimentation are convincing.

weaknesses:
- improvements in quality are minimal. The main benefit is training efficiency for unsupervised MT, which has lower significance than demonstrating quality improvements, or applicability beyond unsupervised MT.